# Increasing Sugar Content in Source for Biofuel Production Using Agrochemical and Genetic Approaches at the Stages of BioMass Preharvesting and Harvesting

**DOI:** 10.3390/molecules27165210

**Published:** 2022-08-16

**Authors:** Darya Zolotareva, Alexey Zazybin, Yelizaveta Belyankova, Anuar Dauletbakov, Saniya Tursynbek, Khadichahan Rafikova, Assel Ten, Valentina Yu, Sarah Bayazit, Anna Basharimova, Murat Aydemir

**Affiliations:** 1School of Chemical Engineering, Kazakh-British Technical University, Tole bi Street 59, Almaty 050000, Kazakhstan; 2Department of Chemical and Biochemical Engineering, Institute of Oil and Gas Geology, Satbayev University, Almaty 050013, Kazakhstan; 3Laboratory of Synthetic and Natural Medicinal Compounds Chemistry, A. B. Bekturov Institute of Chemical Sciences, Sh. Ualikhanov, Almaty 050010, Kazakhstan; 4Department of Chemistry, Dicle University, Diyarbakır 21280, Turkey

**Keywords:** sugar content, bioethanol production, biomass valorization, phytohormones, ripening agents, accumulation of carbohydrates

## Abstract

In order to optimize biofuel (including bioethanol) production processes, various problems need to be solved, such as increasing the sugar content of raw materials/biomass to gain a higher yield of the product. This task can be solved in several ways, with their own advantages and disadvantages, and an integrated approach, such as using a combination of ripening agents and phytohormones or application of a superabsorbent polymer with at least one sugar-enhancing agent, can be applied as well. Here, we reviewed several methods, including pre- and postharvest factors (light, temperature, partial replacement of potassium with magnesium, etc.), genetic modifications (traditional breeding, phytohormones, etc.), chemical ripening methods (Ethephon, Moddus, etc.), and some alternative methods (DMSO treatment, ionic liquids, etc.). The aim of this review was to provide a comprehensive, up-to-date summary of methods of increasing the carbohydrate level in plants/biomass for bioethanol production.

## 1. Introduction

Fossil fuels are non-renewable resources that will be depleted in whole or in part in the foreseeable future. In addition, fossil fuels are the cause of environmental disasters, such as pollution of the atmosphere, the greenhouse effect, acid rain, soil and water pollution, climate change, relief disruption, the destruction of habitats of flora and fauna, the drainage and waterlogging of lands, and the drying up of water bodies [1]. For these reasons, there is a search for the most efficient green sources of fuel and energy. There are ways to improve the quality of conventional fuels; for example, fuel can be purified by desulfurization and denitrogenation, which significantly reduces the emission of harmful substances into the atmosphere [2]. However, renewable energy is, without a doubt, the future of our green planet [3]. It is precisely for bioenergetics that raw materials are needed in the form of biological waste. In this review article, we aimed to look, in detail, at the methods and factors helping to increase the content of sugars in plants, biomass, or raw materials to obtain high yields for further production of biofuels. Among other issues, the introduction of methods that foster the accumulation of sugars in plants will allow us to obtain more valuable yields for food and forage.

To grow, develop, withstand stress, and reproduce, a plant needs energy, which it receives using stored carbohydrates (or sugars). Sugars are classified into monosaccharides (galactose, glucose, fructose), disaccharides (sucrose, maltose, lactose), oligosaccharides (fructans, raffinose family oligosaccharides), and polysaccharides (starch, glycogen, cellulose) [4]. In nature, free monosaccharides, except for D-glucose and D-fructose, are rare. Monosaccharides contain hydroxyl groups and an aldehyde (aldoses) or keto group (ketoses). Oligosaccharides are carbohydrates containing 2 to 10 monosaccharide residues. Disaccharides are a special case of oligosaccharides. Polysaccharides are polymers of monosaccharides (glycans). Typically, plants accumulate sugars in mild stress conditions [4]; we consider these conditions below.

### Raw Materials Used in the Production of Bioethanol

An increase in the sugar content leads to an increase in the yield of bioethanol and a decrease in the cost of its production [5]. It is important to pretreat biomass/raw materials to produce the maximum amount of sugar for further production of biofuels or chemicals. There are four generations of raw materials [1] (Table 1). The methods of pretreatment of biomass include dilute acid hydrolysis, steam explosion, alkaline hydrolysis, ammonia treatment, and hydrothermal, organosolv, and biological processes for lignocellulosics [6]. To date, dilute acid pretreatment is considered the most profitable and cost-effective [1].

The need for green energy sources, as well as the desire to improve the quality of crops for various needs, has led to the search for methods to increase the level of carbohydrates in plants. The difficulty lies in the fact that the internal mechanisms of plants for sugar accumulation are not fully understood and therefore represent a huge field for research.

In comparison with previously published reviews [19] on this issue, this review presents the latest works on this topic over the past 3 years and pays attention to methods that were not mentioned in earlier reviews. Previously published reviews and articles [20,21,22,23,24] focused on the pretreatment of raw materials to facilitate the further processing of lignocellulose, and not on the natural increase in the sugar content during the growing process, which is the focus of this review. The purpose of this review was to help provide an idea of how the application of simple agricultural practices, in combination with the treatment of plants with chemicals that can stimulate an increase in the sugar content of the crop, will help increase the valorization of biomass, and to show directions for the search for new substances that promote the accumulation of sugars in plants.

## 2. Methods of Increasing Sugar Level

Methods for increasing the level of sugars in plants can be conditionally divided into environmental factors, genetic modifications, chemical ripening methods, and alternative methods (Table 2). In turn, in genetic modifications, it is especially necessary to highlight the effect of phytohormones. All the methods stated in Table 2 should be applied with care. For example, Roundup was classified by the WHO as a probable human carcinogen [25], some phytohormones may have a negative effect on male physiology by reducing sperm motility [26], and DMSO causes cardiovascular and respiratory adverse reactions [27]. Nonetheless, the effects are often transient and mild and do not qualify as significant adverse events.

Below, we consider these methods and their features in more detail.

### 2.1. Pre- and Postharvest Factors

Temperature, light, nutrition, watering, mechanical stress, storage conditions, and crop maturity have a great influence on the sugar content (Figure 1).

Light. Photosynthesis is directly related to light because this process takes place only in the light. Photosynthesis is a complex chemical process of converting the energy of visible light (in some cases, infrared radiation) into the energy of chemical bonds of organic substances with the participation of photosynthetic pigments (chlorophyll in plants). In short photoperiods, plants accumulate sugars faster during the light period (day) and degrade them more slowly during the nighttime [28]. The influence of red and blue light on the sugar level was investigated by Chen [29]. In the treatment of lettuce carried out with red light (wavelength, 660 nm) and blue light (wavelength, 450 nm) in different modes (monochromatic red or blue, simultaneous red and blue, mixed modes of monochromatic and simultaneous light, alternating red and blue with an interval of 4 h, and alternating red and blue with an interval of 1 h), the best result was obtained under the mode of alternating red and blue light with an interval of 1 h. This mode caused the best accumulation of starch, sucrose, and biomass in the lettuce [28].

Carbon dioxide. Among other things, carbon dioxide also has a positive effect on photosynthesis in plants, which means an increase in the sugar level in agricultural crops [30]. The research of Zheng [31] on soybeans revealed a correlation between the sugar level and the concentration of carbon dioxide in the atmosphere. The concentration of soluble sugars and starch increased with the concentration of CO_2_ up to 800 ppm, but at an extremely high CO_2_ concentration (1200 ppm), the level of carbohydrates fell sharply. Starch is one of the main carbohydrates in plants, the content of which increases with increasing carbon dioxide concentration [32]. Further, starch can be converted into sucrose. Higher CO_2_ levels affect root growth as the concentration of sucrose increases because of the increasing photosynthetic rate. CO_2_ enhancement increases both the total number of roots and their length in A. thaliana, and the diameter of the roots [33].

Temperature affects photosynthesis, which is inextricably linked to the formation of carbohydrates (i.e., sugars) [34]. It has been proven by Mathieu et al. [35] that high temperatures negatively affect the sugar level in chicory (*C. intybus*). Even a 5 °C increase in the overall temperature during the growth season causes a decrease in the sugar level in chicory. On the other side, ref. [36] showed that a lower temperature (about 3 °C) during the daytime and nighttime enhanced the sucrose concentration in sugarcane. Recent research [37] showed that during plant senescence at high temperatures, concentrations of sugars lowered, while they increased at low temperatures, even at 12 °C. Extremal temperatures, of course, are very harmful to crops [38]. Carbohydrate accumulation noticeably reduces under heat stress, because the respiration rate increases under high temperatures (above 30 °C), which leads to a decrease in the photosynthetic rate [39].

Soil nutrition also has an impact on the sugar content. Adequate use of N, P_2_O_5_, and K_2_O fertilizers achieves an increase in sugar levels in sweet sorghum; moreover, the nitrogen amount plays a more substantial role than the amounts of other fertilizers [40]. Morrow et al. [41] also proved the efficiency of N fertilizers in increasing the sugar level in orange trees; however, on the other side, the use of fertilizers decreased the normal microbial biomass of the soil, which negatively affects the ecosystem. In ref. [42], Barłóg et al. developed recommendations for the use of K, Na, and Mg fertilizers to improve beet yield, quality of root crops during storage, and sugar yield. The dependence of the ratio of these elements, the influence of the partial replacement of potassium (K) with magnesium, the composition of the soil, and climatic conditions on the listed indicators were shown. It turned out that a ratio of K:Na in fertilizers that is too narrow leads to a decrease in the sugar content in root crops; in addition, an excess of available potassium in the soil affects the distribution of carbohydrates between the tops and roots, which leads to a decrease in the sugar content of the root crop.

Some herbicides increase the sucrose content and sucrose yield. Maximum consumption of sugar is due to the overgrowth that occurs shortly before harvest. The process is partially controlled by reducing watering, as well as limiting the application of nitrogen fertilizers during this period. It turns out that these methods of plant maturation are unreliable and difficult to control. The use of sulfonylurea derivatives, known for their herbicidal action, has been proposed, which suppress the vegetative growth of the plant in the period 2–8 weeks after full maturation. The effectiveness of this method [43] has been shown on sugarcane and sorghum in greenhouse and field trials. The use of sulfonylurea herbicides 2–8 weeks after the plant is fully matured makes it possible to stop the “expenditure” of sugar for its growth and redirects carbohydrate metabolism in such a way as to increase the level of sugar in the plant juice. The herbicides Plateen 41.5WG and Racer 250 EC applied to control weeds in potato increased total sugars in potato tubers [44]. Maleic hydrazide (Fazor) has an effect on the sucrose content (best treatment shows a 1.05% increase) and sucrose yield (0.3% treatment shows an increase of 6.2% in sucrose yield per acre) of sugar beet [45]. Maleic hydrazides increase the tuber sucrose and glucose concentration at harvest in Atlantic potato tubers [46]. Exposing the rice seedling Jinyou 402 to combined treatment of cadmium with acetochlor and bensulfuronmethyl improved the soluble sugar content by 2.84% and 1.23%, respectively [47]. Moreover, even herbicide safeners (for example, cloquintocet-mexyl, quinoline’s derivatives) could be used for increasing the sugar content and biomass yield of sugar plants [48].

The fungicide Kagatnik not only stops the rotting of the obtained raw materials but also increases the sugar content of sugar beet roots. Kagatnik was introduced during the growing season, and it contributed to more intensive outflow of nutrients from the leaves to root crops, due to which there was a sharp increase in the sugar content [49].

Abiotic stress: osmotic stress and heavy metal poisoning. Osmotic stress induces the accumulation of soluble sugars (fructose, glucose, sucrose, maltose, and galactose) [50] in wheat treated with PEG-6000, mannitol, sorbitol, or NaCl. More precisely, PEG treatment causes the accumulation of glucose and fructose both in roots and in leaves. Mannitol and sorbitol treatments cause a decrease in the RWC, chlorophyll, and photosynthetic activity of the leaves. NaCl treatment causes the accumulation of proline, sucrose, and galactose. Total soluble carbohydrates increase in rice under salt stress [51]. On the other side, the starch level decreases under salt stress [52], perhaps because the salt stress has an inhibitory effect on photosynthesis [53]. Remarkable enhancement of sucrose, glucose, and fructose levels was observed at high concentrations of salt, because they participate in carbon storage, osmoprotection, osmotic homeostasis, and neutralization of free radicals [54]. The high concentration of copper and nickel inhibits the process of photosynthesis, which leads to a lower sugar content in *Glycine max* [55]. Heavy metals (Fe, Cu) cause oxidative stress due to reactive oxygen formation and inhibition of photosynthesis and plant metabolism [56]. These reasons contribute to a decrease in the level of carbohydrates in the plant [56].

Biotic stress: fungi, viruses, bacteria, and pest infestation. No matter what pest or organism injures the plant, it will develop with the sugars produced in the host plant [57]. Aphid-infested stems accumulate individual sugars and amino acids, but in infested young leaves, phosphoric acid, arabinose, saccharic acid, and shikimic acid are not detected or are very low [58]. There is an increase in the content of glucose and fructose in grapes infected with the mold fungus *Botrytis cinerea*. This fungus induced a reprogramming of carbohydrate metabolism in grape berries, with a decrease in photosynthesis and an increase in the catabolism of sugars [59]. In ref. [60], the authors proved that root-associated bacterial (Bacillus megaterium and Pseudomonas aeruginosa) treatment leads to carbohydrate accumulation in plants by enhancing photosynthesis.

Postharvest storage is another major factor that changes the sugar content in plants. The sugar level in fruits increases with the storage duration regardless of the temperature and treatment after harvest [61]. In ref. [62], fruits were treated with CaCl_2_ and hexanal at different temperatures (18 ± 2 and 28 ± 2 °C). According to the results of the research, hexanal-treated fruits had higher sugar levels than calcium chloride-treated or untreated fruits. Moreover, total sugars and reducing sugars increased over the storage duration.

### 2.2. Genetic Modifications

The science of biotechnology offers its own methods for increasing the level of sugars in plants. These methods include the genetic modification of photosynthetic pathways, the cell architecture, and phytohormones [59] (Figure 2). Lima et al. [59] presented an extended table of many genes that are involved in the control of growth and productivity.

In ref. [63], it was shown that well-planned breeding schemes can lead to an improvement in sorghum sugar characteristics such as the sugar content and biomass. An improvement of sugarcane in terms of biomass has been achieved by traditional breeding, molecular genetic approaches, and NGS (next-generation sequencing) technology, resulting in a biomass with better digestibility, modified carbohydrates, and a reduction in cross-linking or self-produced enzymes (in planta) [64]. There has been some success in increasing the total sugar content in sugarcane by introducing a bacterial sucrose isomerase gene through genetic modification [19].

Growth regulators, called plant hormones, including abscisic acid (ABA), ethylene, gibberellins (GAs), auxin (IAA), cytokinins, and brassinosteroids (BRs) can exert strong action on physiological and biochemical processes in plants [65] (Figure 2). Sugars and phytohormones are individually viewed as major players in many aspects of plant biology. Their crosstalk has not been systematically investigated, and hence many gaps in the current knowledge exist. Moreover, the current research underlines that the crosstalk is very complex and varies at least according to the nature of the organ and the physiological process.

Abscisic acid (ABA) is a vital central regulator of plant stress such as drought, salinity, low temperatures, and osmotic stress [66,67], which, in turn, leads to the accumulation of sugars in plants. In [68], the synergy between ABA and sucrose was reported. In response to the ABA plus sucrose treatment, accumulation of carbohydrates in rice was observed, which is a consequence of the large improvement in sucrose transport. ABA has stimulatory action on glucose metabolism [69]. It is indicated that sugar accumulation in peach flesh is effectively stimulated by ABA both in vivo and in vitro [70]. The same conclusion of an effect of ABA on sugar accumulation in sweet sorghum was drawn in ref. [70], and it was assumed that this effect was due to genes involved in sugar metabolism and transport.

The auxin (IIA) concentration has been reduced by glucose in the roots of Arabidopsis due to a reduction in PIN1 protein levels [69]. Sugars, especially glucose, can regulate the rate-limiting step in auxin biosynthesis, and the availability of sugars may change the synthesis of auxin biosynthetic enzymes and thus regulate cell division and sink size based on these signals [71]. Bud outgrowth quantitatively adjusts to the balance between sugar and auxin levels, with increased sugar leading to a strong reduction in bud inhibition by auxin; the sugar effect involves repression of the strigolactone response [72].

Cytokinin (CK) and glucose are found to behave antagonistically at lower concentrations and agonistically at higher concentrations [69]. CK activity is independent of photoreceptors but highly dependent on the redox state of the photosynthetic electron transport chain whereby the redox poise regulates the pigment biosynthesis [73]. Based on the results derived from different plant species, sugars and CKs seem to act synergistically to take over the seedling emergence, shoot meristem activity, and shoot branching and flowering, while they act antagonistically in seed germination, root meristematic activity, and root branching and leaf senescence [74]. In many plant species, cytokinins positively affect photosynthetic rates, which is associated with increases in stomatal conductance and gas exchange, leading to higher photosynthetic rates and sucrose production [75]. Cytokinins may also play a role in sucrose transport from source to sink organs by regulating the expression of SWEET and SUT/SUC transporters [75].

Glucose and ethylene have been shown to be antagonistic in their signaling pathways [69]. Ethylene regulates the process of photosynthesis by reducing glucose sensitivity [69]. Ethylene treatment increased the stem sucrose content, but that occurred only in a low-sugar genotype of sugarcane. Sucrose and starch metabolism genes were more responsive to ethylene treatment in a low-sugar genotype [76]. The rapid decline in sugars in broccoli florets after harvest may influence the potential of endogenous ethylene function that involves chlorophyll degradation, autocatalytic ethylene production, and related ethylene-induced senescence [77]. With sucrose feeding ethylene production, the ACC (1-aminocyclopropane-1-carboxylic acid) content and ACS (1-aminocyclopropane-1-carboxylic synthase) activity increased significantly in florets [77]. Exogeneous ethylene treatment showed an increase in the content of reducing and non-reducing sugars (starch and polysaccharides) in Alphonso mango [78]. Although it remains unclear why sugars induce anthocyanin and ethylene accumulation simultaneously, it is worth noting that the ability of ethylene to repress anthocyanin accumulation varies with respect to the light intensity and sugar concentration [79].

Brassinosteroids (BRs) are phytohormones that have positive effects in stress conditions; they regulate genes involved in various key processes in plants such as photomorphogenesis, flowering, and biotic and abiotic stress responses [80]. Glucose and BRs act antagonistically at low glucose concentrations and synergistically at higher glucose concentrations in hypocotyl elongation growth regulation in dark-grown seedlings [81,82]. In ref. [83], it was indicated that light modulates the sugar–BR crosstalk. Sugar increases BR hormone accumulation in the dark but decreases the BR level under light [80]. Recently, several reports [84,85] showed that light signaling inhibits BR signaling through photoreceptors. BR biosynthesis and function are required for carbon uptake and carbohydrate metabolism, affecting the efficiency of nutrient exchange between both symbionts and the mycorrhizal growth benefit for the plant [85].

### 2.3. Chemical Ripening Methods

Chemical ripening agents (Figure 3) are widely used to accelerate the ripening process and could be applied prior to harvest for increasing the sucrose concentration in crops as well. Although chemical ripeners may reduce photosynthetic rates (on a leaf area basis), their chemical inhibition of new leaf growth has a much greater effect on increasing sucrose accumulation by reducing growth sink requirements for sucrose [86].

Ethephon™ (2-chloroethylphosphonic acid) has a hormonal mechanism of action, and it is effective due to ethylene release [87]. The effects of ethephon on sugarcane include a higher accumulation of sucrose, higher biomass production, and a higher sugar yield under diverse agroclimatic conditions [87,88]. Moreover, treatment of sugarcane with a combination of ripening agents (ethephon) and phytohormones (GA) improved the sucrose content (by 1.05%) and yield of commercial cane sugar (13.23 t/ha) [89]. A recent study [90] showed that ethephon treatment of the Rasi and Taipei-309 rice cultivars increased the soluble sugar content by 12–14.8% (under non-stress conditions) and by 40.1–45.6% (under high temperature) compared with the control.

Moddus™ (trinexapac-ethyl) is a cyclohexanedione growth regulator with a hormonal mechanism, which inhibits the production of GA and leads to the restriction of internode elongation [91]. According to ref. [92], trinexapac-ethyl treatment of sweet sorghum (KKU 40 cultivar) caused the highest growth and sugar yield when applied at 0.05 ppm and preharvest week 1.

Roundup™ (glyphosate) and Fusilade Forte™ (fluazifop-p-butyl) are herbicides that terminate new tissue formation at sub-lethal doses; however, at very low concentrations, they are effective chemical ripeners [86]. Glyphosate is used to significantly increase sugar accumulation and improve the sugar yield in sugarcane [93,94]. The application [95] of Fusilade Forte™ resulted in the highest sugar yield, which was 35.6% higher compared with the control; however, it showed a reduction in the sugarcane yield. A study of 43 Australian sugarcane genotypes showed an increase in sucrose after application of glyphosate and combined application of ethephon and Fusilade Forte™ [96].

Although combinations of chemical ripeners with other ripeners and/or plant growth stimulators lead to significant results in terms of the sugar content, sugar yield, and biomass production, the response of different genotypes of plants to the ripening treatment may vary and depends on many other factors [86,87,89,96,97].

### 2.4. Alternative Methods

One of the “alternative” methods (Figure 4) to increase the sugar content in sugarcane crops is the usage of a superabsorbent polymer (SAP) with at least one sugar-enhancing agent or herbicide [98]. A study [99] of the effectiveness of an SAP with less than half the amount of fertilizer (compared with the control plant) demonstrated an increase in biomass accumulation and the sugar content. SAP treatment of the Kosmas and Brian varieties of sugar beets in dry and warm regions has shown a significant effect on the white sugar yield and a positive influence on the sugar content [100].

DMSO treatment improves the sugar content by 0.2–1.2% [101], and DMSO/titanium tetrachloride (1.5% DMSO + 0.05% Ti) solution spraying during the leaf formation of sugar beets and 10–15 days before harvest improves the sugar content by 0.4–1.9% [102]. However, it is evident that DMSO accumulates a water body without any indication of its presence due to its high polarity through anthropogenic input [103], and that it has toxic effects due to its high osmolarity [104]. Rice seedlings exposed to DMSO for 72 h showed an increase in soluble sugars, although the relative growth length, water use efficiency, chlorophyll, and protein content significantly decreased, resulting in growth inhibition and cell death [105].

By increasing the concentration of imidazole ionic liquids in rice (Hangzhou Liangyouxiangzhan) and capsicum (Changshun Prince), the content of the reducing sugar in root cells increased significantly with the destruction of the cell membrane [106]; however, imidazolium-based ionic liquids are well known for their harmful toxicity toward aquatic organisms [107] and poor biodegradability [108]. Nevertheless, due to their properties, ionic liquids are used for biomass pretreatment in order to generate a high glucose yield [109]. The efficiency of ionic liquid pretreatment is proved by its ability to improve cellulose accessibility and increase sugars overall [110,111]. For biomass pretreatment in biofuel production, ionic liquids are proposed as an environmentally friendly method. According to the results of experiments, 1,3-dimethyl-imidazolium methyl phosphonate is a more efficient solvent in the pretreatment of miscanthus than other ionic liquids. The high hydrogen bond basicity and polarity compared with most ionic liquids are presumably the reasons for this dissolution efficiency. During biofuel production, the formation of porous cellulose occurs with a low percentage of lignin, which facilitates its enzymatic hydrolysis. The overall efficiency of glucose hydrolysis for cellulose after regeneration can reach up to 97%. Another example of the effective influence of ionic liquids in increasing sugars can be observed in a study by Socha et al. [112]. Due to the high costs and therefore the limitations of industrial-scale deployment of imidazolium cations, the researchers generated the synthesis of tertiary amine-based ionic liquids using aromatic aldehydes derived from hemicellulose lignin, the major byproduct of lignocellulosic biofuel production. After a 72 h incubation, 90–95% glucose and 70–75% xylose yields were obtained from samples of switchgrass pretreated with ionic liquids derived from biomass. Although these results are lower than those for samples pretreated with 1-ethyl-3-methylimidazolium acetate, they are still efficient and also confirm the prospect of creating a cycled process and decreasing the cost of ionic liquids which can be potentially used as a pretreatment.

Metal ions can be used as enzyme effectors. Mg^2+^ and Mn^2+^ ions improve the sucrose content and increase sucrose synthesis in low-sucrose-accumulating genotypes of sugarcane, which might be due to a change in the soluble acid invertase activity pattern of sucrose-synthesizing enzymes, which also helps to increase the commercial cane yield [113].

## 3. Conclusions

There are some exogenous factors that affect the level of sugars in plants, and that can be applied to increase the yield of harvest or improve its quality, for example, the use of red and blue light, an increase in the length of daylight hours, or a slight decrease in the ambient temperature. Among the other methods, it is possible to mention the treatment of the crop with carbon dioxide or hexanal at certain concentrations, or adequate fertilization and watering of the soil. The absence of pests can also lead to the accumulation of carbohydrates in the plant/biomass, and the potential for using chemical activators to increase sugar levels in crops is far from being exhausted. However, even more complex research lies in the field of genetic engineering and selection. Through genetic research, one can see the cross-effects of phytohormones and carbohydrates, which means adjusting the necessary properties to obtain the best result. Chemical ripening and alternative methods are also widely used to increase the sugar content in plants/biomass. Therefore, the choice of methods or their combined application will make it possible to achieve significant success in increasing the sugar content of raw materials to achieve the highest yield of bioethanol.

The authors believe that practically each of the pre- and post-harvest methods can be applied in practice, since they are quite simple agricultural methods that do not require serious investments and technologies. Regarding genetic modification methods, including the use of phytohormones, the mechanisms behind these methods are not fully understood, but they represent the most promising direction. Chemical ripening methods are well established and can be used in combination with other methods or for multiple purposes, such as increasing biomass and as herbicides. Alternative methods are not well understood, but the authors hope that the scientific community will pay attention to them, especially to ionic liquids due to their special properties and the ability to fine-tune.

## Figures and Tables

**Figure 1 molecules-27-05210-f001:**
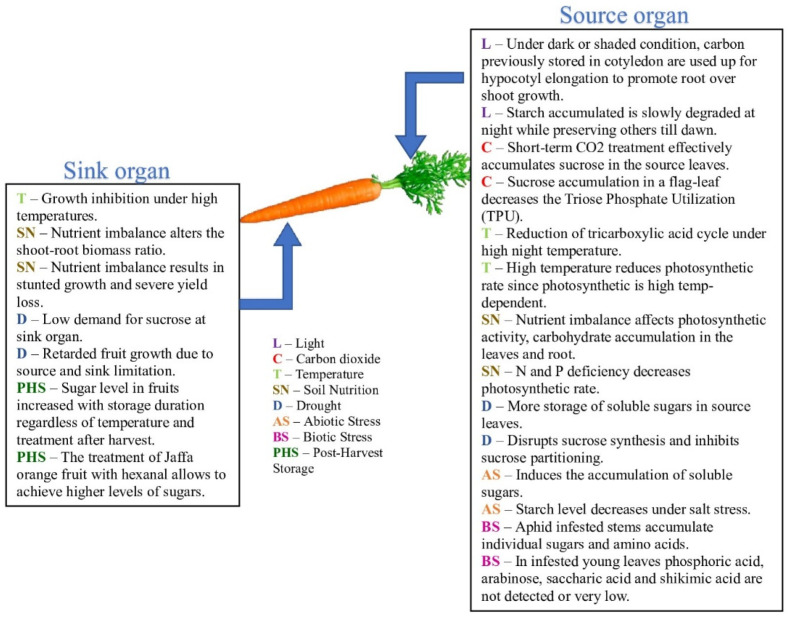
The influence of different factors on the sugar concentration.

**Figure 2 molecules-27-05210-f002:**
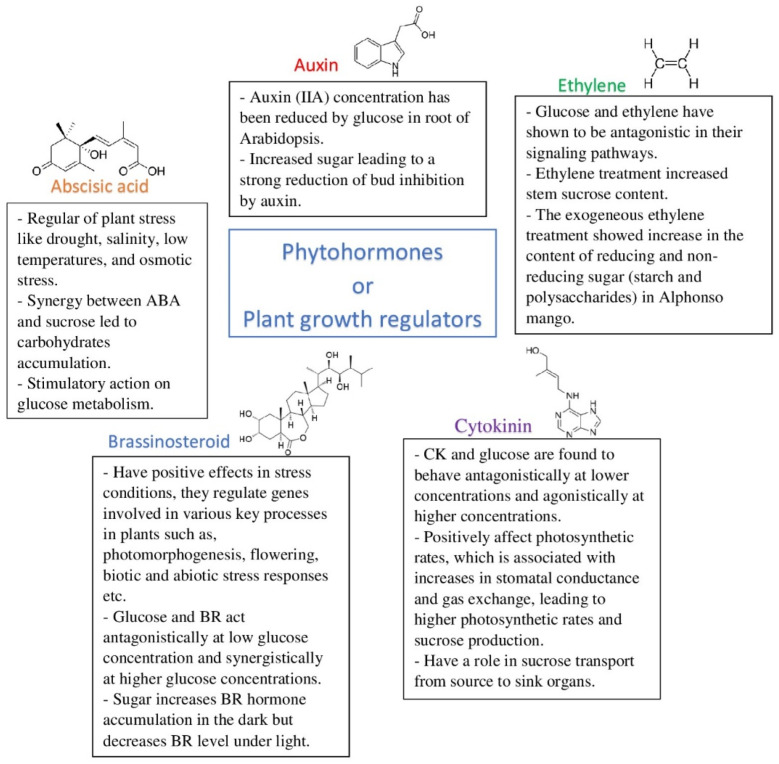
The effect of phytohormones on plants and the sugar level.

**Figure 3 molecules-27-05210-f003:**
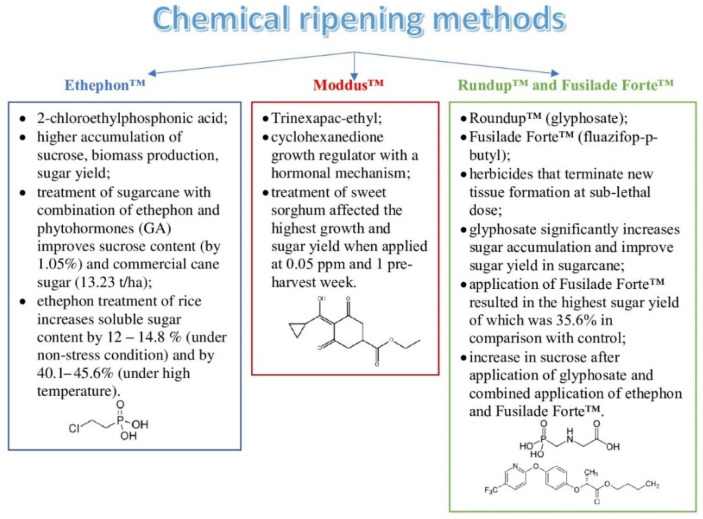
Chemical ripening methods.

**Figure 4 molecules-27-05210-f004:**
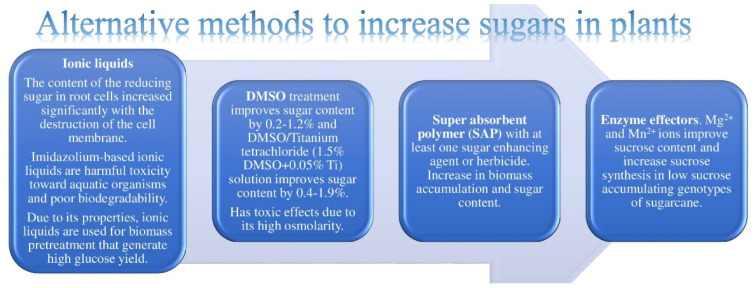
Alternative methods to increase sugars in plants/biomass.

**Table 1 molecules-27-05210-t001:** Raw materials used in the production of ethanol.

Generation	Pretreatment Process	Raw Material
First (1G)	Need only milling, fermentation, distillation, and denaturalization (in case of human consumption). For use in mixtures with gasoline, the material must be dehydrated [7]. Fermentation by microorganisms *Saccharomyces, Zymomonas, Kluyveromyces,* and *Zygosaccharomyces* [8].	Sugarcane, sugar beets, and sweet sorghum [9], high-starch content plants (cereals, tubers, and rhizomes) [10].
Second (2G)	Pretreatments, enzyme hydrolysis and fermentation [7,11].	Farm residue (cereal straw, leaves, dry branches of forest crops) or industrial residues (sugarcane bagasse and DDGS (distillers’ dried grain with solubles)) [7].
Third (3G)	Pretreatments, enzyme hydrolysis, and fermentation [12].	Perennial grasses, micro- and macro-algae, and cyanobacteria [13].
Fourth (4G)	No-destruction of biomass, direct conversion of solar energy to fuel [14]. Acid/enzymatic hydrolysis, fermentation [15,16].	Genetically or metabolically modified organisms (GMO), for example, GM sugarcane [17], algae [18].

**Table 2 molecules-27-05210-t002:** Methods to increase the sugar level in plants.

№	Method	Short Description
1	Pre- and postharvest factors	Light
		Carbon dioxide
		Temperature
		Soil nutrition
		Abiotic stress
		Biotic stress
		Postharvest storage
2	Genetic modifications	Traditional breeding
		Molecular genetic approaches
		NGS (next-generation sequencing) technology
		Phytohormones
3	Chemical ripening methods	Ethephon™ (2-chloroethylphosphonic acid)
		Moddus™ (Trinexapac-ethyl)
		Roundup™ (glyphosate)
		Fusilade Forte™ (fluazifop-p-butyl)
4	Alternative methods	Superabsorbent polymer (SAP)
		DMSO
		DMSO/titanium tetrachloride
		Ionic liquids
		Enzyme effectors

## Data Availability

Not applicable.

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
