# Peer review of "Increasing Sugar Content in Source for Biofuel Production Using Agrochemical and Genetic Approaches at the Stages of BioMass Preharvesting and Harvesting"

_molecules, 2022, doi:10.3390/molecules27165210_

Round 1

Reviewer 1 Report

The article molecules-1853375 showed improvements, including the English language revised.

However, table 1 still contains errors. For example, correct for 2G “Pretreatments, enzyme hydrolysis and fermentation”;

- It is necessary to clarify whether for 4G feedstock pretreatment and enzymatic hydrolysis are needed;

- I suggest reading and adding two new references, which will help solve the problems still existing in this table:

https://link.springer.com/chapter/10.1007/978-981-13-8637-4_7

https://doi.org/10.1016/j.esd.2022.03.007

Author Response

The article molecules-1853375 showed improvements, including the English language revised.

However, table 1 still contains errors. For example, correct for 2G “Pretreatments, enzyme hydrolysis and fermentation”;

Answer: Done

- It is necessary to clarify whether for 4G feedstock pretreatment and enzymatic hydrolysis are needed;

Answer: There is no need of pretreatment of the biomass for 4G. We fixed it. Thank you. However, the references you gave below claim that 4G fuels need acid/enzyme hydrolysis and fermentation, so, we added both.

- I suggest reading and adding two new references, which will help solve the problems still existing in this table:

https://link.springer.com/chapter/10.1007/978-981-13-8637-4_7

https://doi.org/10.1016/j.esd.2022.03.007

Answer: The references were taken into consideration and were added to the list.

Reviewer 2 Report

The authors review several methods to provide a comprehensive, up-to-date summary of methods of increasing the carbohydrate level in plants/biomass for bioethanol production.. Some recommendations are put forward for improvement, as follows:

(1) There are some revision formats in the context, Page 2, Line 65, Table 1, for examples;

(2)Page 2, Line 46, "What are sugars? ", the expression is not very academic;

(3)Page 6, Line 213, "In [60]", generally, we don't express as this, we prefer to use "In xxx's study", or "in reference [60]";

(4)In Abstract, "In order to optimize biofuel production processes, including bioethanol......", does bioethanol belongs to "optimize biofuel production processes"?

(5) Table2, there is no reference marks;

(6) Figure1 (Page6) is drawn by the authors or cited from reference? It's clarity needs to be improved;

(7) Some mechanistic explanations for various methods are needed, not just a simple list some experimental results;

(8)The crosswise comparison for methods to increase the sugar level in plants (as shown in Table 2) is suggested to be added;

(9)What are the applicable scenarios for the different methods to increase the sugar level in plants (as shown in Table 2);

(10) In conclusion part, some outlook perspective sentences need to be added.

Besides, some sentences desecription needs to be revised by a English native speaker.

Author Response

  •  

(2)Page 2, Line 46, "What are sugars? ", the expression is not very academic;

Answer: This sentence was deleted.

(3)Page 6, Line 213, "In [60]", generally, we don't express as this, we prefer to use "In xxx's study", or "in reference [60]";

Answer: Fixed.

(4)In Abstract, "In order to optimize biofuel production processes, including bioethanol......", does bioethanol belongs to "optimize biofuel production processes"?

Answer: Yes, bioethanol belongs to optimize biofuel production processes". IN order to make this sentence clearer we changed it to “In order to optimize biofuel (including bioethanol) production processes, various”

(5) Table2, there is no reference marks;

Answer: Authors think that there is no need for the reference marks in Table 2, since the table is for informational purposes only and indicates what we will consider in the text below.

(6) Figure1 (Page6) is drawn by the authors or cited from reference? It's clarity needs to be improved;

Answer: The idea for the Figure 1 was taken from the reference 32 [Figure 1 in 35à32, Aluko et al.] and was significantly modified in accordance with the content of this review. We have improved the text quality as well.

(7) Some mechanistic explanations for various methods are needed, not just a simple list some experimental results;

Answer: In this review, we give the explanation for the mechanism of action for sugar content increasing (see, for example, lines 189, 235, 239, 258, etc.), but only in the case it was stated or suggested by the authors quoted. We also give some examples without an explanation of the mechanism of action, where the mechanism is unknown, but an example is illustrative.

(8)The crosswise comparison for methods to increase the sugar level in plants (as shown in Table 2) is suggested to be added;

Answer: In this review, we provide information about different approaches for increasing the sugar content in plants at pre-harvesting and harvesting, first dividing these methods into several groups (Table 2) and thereafter providing specific examples for each group of methods. The crosswise comparison is difficult because examples in different articles are given for different crops and growing conditions. But, in conclusion, we mark the most promising methods.

(9)What are the applicable scenarios for the different methods to increase the sugar level in plants (as shown in Table 2);

Answer: The applicable scenarios were added in Conclusion section.

(10) In conclusion part, some outlook perspective sentences need to be added.

Answer: See the answer below.

Besides, some sentences desecription needs to be revised by a English native speaker.

Answer: English was checked by MDPI native speaker (we have a certificate).

Round 2

Reviewer 2 Report

The authors have addressed all of my concerns and and can be acceptted in the present form.

This manuscript is a resubmission of an earlier submission. The following is a list of the peer review reports and author responses from that submission.

Round 1

Reviewer 1 Report

1. Abstract should be more elaborative.

2. Keywords should be more specific.

3. Introduction required improvement 

4. Add more studies on biotic and abiotic stress factors. 

5. Figures are not clear.

6. Discuss the uniqueness of this review.

Author Response

  1. Abstract should be more elaborative.

Answer: The abstract has been expanded.

  1. Keywords should be more specific.

Answer: The keywords were updated.

  1. Introduction required improvement 

Answer: The introduction was expanded

  1. Add more studies on biotic and abiotic stress factors. 

Answer: Added.

  1. Figures are not clear.

Answer:  Figure 1 illustrates the influence of a range of agricultural techniques on sink–to-sourse transport – the important process for accumulating sugars in biomass (e.g. transferring of feed substances from the sugar cane root to the leaves and stalk).

Figure 2 demonstrates the brief information of phytohormones effect on sugar content in plants and sugar-phytohormone mutual influence.

Figure 3 also demonstrates brief information of chemical ripening methods and their effect on sugar content.

Figure 4 demonstrates brief information of alternative methods as well for easy visual comparison of these ones.

  1. Discuss the uniqueness of this review.

Answer: The uniqueness of our review displayed in the added last paragraph of Introduction.

Reviewer 2 Report

The review article molecules-1794701 needs to highlight what is novelty in comparison to other review articles already published on this subject. It is clear in the text that the authors do not master the fundamentals related to pretreatments of biomass for conversion into ethanol. Table 1 has many errors.

- Line 57-60: Specify that these treatments are for lignocellulosics;

- Table 1: Change to “enzyme hydrolysis”;

- Table 1: Lignin is not a saccharide;

- Table 1: Pretreatments and fermentation for 2G are missing;

- Table 1: Thermochemical treatments can also be applied to algae;

- Table 1: For algal biomass there is also a need for pretreatments and enzymatic hydrolysis; the same applies for 4G;

- Line 86: The authors must cite only examples of plants destined for ethanol production;

- Line 168-169: Rephrase this sentence;

- It is also important to discuss the adverse effects of the various inputs mentioned in Table 2.

Author Response

  1. The review article molecules-1794701needs to highlight what is novelty in comparison to other review articles already published on this subject. It is clear in the text that the authors do not master the fundamentals related to pretreatments of biomass for conversion into ethanol.

Answer: The novelty of our review was added in last paragraph of Introduction.

Table 1 has many errors.

  1. - Line 57-60: Specify that these treatments are for lignocellulosics;

Answer: Added.

  1. - Table 1: Change to “enzyme hydrolysis”;

Answer: Changed.

  1. - Table 1: Lignin is not a saccharide;

Answer: Deleted.

  1. - Table 1: Pretreatments and fermentation for 2G are missing;

Answer: Added information.

  1. - Table 1: Thermochemical treatments can also be applied to algae;

Answer: Added.

  1. - Table 1: For algal biomass there is also a need for pretreatments and enzymatic hydrolysis; the same applies for 4G;

Answer: Added.

  1. - Line 86: The authors must cite only examples of plants destined for ethanol production;

Answer: The authors realize that lettuce does not corresponds to the type of biomass used in bioethanol production, but in this work the most diverse investigation of light influence on sugar contest was done and the results may be transferrable to the other biomass types. That is why the authors ask to stay this quotation in the manuscript.

  1. - Line 168-169: Rephrase this sentence;

Answer: Done.

  1. - It is also important to discuss the adverse effects of the various inputs mentioned in Table 2.

Answer: Added above Table 2.

Reviewer 3 Report

This work is entitled Increasing Sugar Yield and Sugar Content in Source For Biofuel Production. One of the main difficulties in the biofuel production process lies in the full use of the sugar content contained in the raw materials, since it is common that, due to the recalcitrance of the cellulose, a compromise must be reached between the availability of sugars for fermentation and the production of inhibitors that hinder conversion into biofuels. With this title, the reader would expect a review of the main methods by which pretreatment can be improved, as a fundamental step to facilitate biomass fractionation, increase the solubilization or hydrolysis of the hemicellulosic fraction, increase the accessibility of enzymes to cellulosic fraction and keep the generation of inhibitors to a minimum.

However, the review is dedicated to describing a series of methods whose impact on improving biofuel generation is debatable.

In the current format, the review does not provide information relevant to the topic.

Some other items to reconsider are outlined below:

- The references to the different renewable energies [3] to [9] are superfluous and are not applicable in a review like this one.

- Table 1. It is the first time that it is said that Fermentation constitutes third generation ethanol production. In addition, the 4th generation category that appears in this table specifically includes two microorganisms, S. shehatae and E. coli, when the number of genetically modified microorganisms is much higher, with numerous application examples. What does the Saccharide column mean? Can lignin be considered a saccharide?

- Table 2. Here a mixture of different methods with an uncertain relationship between them is shown. Many of them, those included in Chemical ripening methods, are reduced to a series of commercial products whose relationship with the increase in sugars can only be justified by their fungicidal or insecticidal nature, which is equivalent to saying that watering the plants would also increase the content of sugars.

IIn Page 4, line 100+, please explain how Co2 can influence the content of sugars

-        Figure 1. Please clarify what the purpose of this figure is and what practical information it adds.

-        Alternative methods. Authors mention ionic liquids, which are a series of compounds used in the pretreatment of biomass for increasing fractionation, e.g. hemicellulose solubilisation.

-        Why not review the methods for improving enzyme accessibility to cellulose fraction for glucose generation, or hemicellulose enzymatic hydrolysis?

Author Response

  1. This work is entitled Increasing Sugar Yield and Sugar Content in Source For Biofuel Production. One of the main difficulties in the biofuel production process lies in the full use of the sugar content contained in the raw materials, since it is common that, due to the recalcitrance of the cellulose, a compromise must be reached between the availability of sugars for fermentation and the production of inhibitors that hinder conversion into biofuels. With this title, the reader would expect a review of the main methods by which pretreatment can be improved, as a fundamental step to facilitate biomass fractionation, increase the solubilization or hydrolysis of the hemicellulosic fraction, increase the accessibility of enzymes to cellulosic fraction and keep the generation of inhibitors to a minimum.

However, the review is dedicated to describing a series of methods whose impact on improving biofuel generation is debatable.

In the current format, the review does not provide information relevant to the topic.

Answer: The authors understand the importance of the research in the area of lignocellulose pretreatment, but consider alternative approaches to biomass valorization as well. Particularly, this review is focused more not on a pretreatment methods but on unorthodox ways for getting available for bioethanol production sugars by means of application of an integrative approach – a combination of agricultural techniques and treatment of plants with chemical compounds capable to facilitate sugars accumulation in biomass.

Some other items to reconsider are outlined below:

  1. - The references to the different renewable energies [3] to [9] are superfluous and are not applicable in a review like this one.

Answer: Changed.

  1. - Table 1. It is the first time that it is said that Fermentation constitutes third generation ethanol production. In addition, the 4th generation category that appears in this table specifically includes two microorganisms, S. shehatae and E. coli, when the number of genetically modified microorganisms is much higher, with numerous application examples. What does the Saccharide column mean? Can lignin be considered a saccharide?

Answer: The table 1 was revised according to recommendations.

  1. - Table 2. Here a mixture of different methods with an uncertain relationship between them is shown. Many of them, those included in Chemical ripening methods, are reduced to a series of commercial products whose relationship with the increase in sugars can only be justified by their fungicidal or insecticidal nature, which is equivalent to saying that watering the plants would also increase the content of sugars.

Answer: Thank you for your comment. In the Table 2 the authors are attempting to make a system, a taxonomy of methods, used for sugar accumulation in plants. The authors agree that there is an evident connection between application of fungicides and insecticides and biomass yield, but there is not obvious relationship between the application of fungicides and insecticides and increasing of sugar content in biomass and additional research is needed to prove whether this or another fungicide or insecticide (and why not all of the fungicides and insecticides?) increases the sugar content in biomass just due to its fungicidal or insecticidal nature or due to some other (e.g. genetic, enzymatic) pathways.

  1. IIn Page 4, line 100+, please explain how Co2 can influence the content of sugars

Answer: Added

  1. -Figure 1. Please clarify what the purpose of this figure is and what practical information it adds.

Answer: Figure 1 illustrates the influence of a range of agricultural techniques on sink–to-sourse transport – the important process for accumulating sugars in biomass (e.g. transferring of feed substances from the sugar cane root to the leaves and stalk).

  1. -        Alternative methods. Authors mention ionic liquids, which are a series of compounds used in the pretreatment of biomass for increasing fractionation, e.g. hemicellulose solubilisation.

Answer: Ionic liquids are mentioned in this review as potent compounds to force increase in sugars. These days mainly the view on ionic liquids is as on the promising agents for biomass pretreatment (as a media). But ionic liquids due to their ability to easily penetrate through the plant cells and influence the enzymatic system may cause (in small concentrations) the increase in sugar as well.

  1. -        Why not review the methods for improving enzyme accessibility to cellulose fraction for glucose generation, or hemicellulose enzymatic hydrolysis?

Answer: Thank you for the advance, the authors understand the importance of the reviews on the methods for improving enzyme accessibility to cellulose fraction for glucose generation, or hemicellulose enzymatic hydrolysis, but the numerus reviews on these and similar topics are recently published (надо оформить ссылки по правилам):

1) da Silva, A.S., Espinheira, R.P., Teixeira, R.S.S. et al. Constraints and advances in high-solids enzymatic hydrolysis of lignocellulosic biomass: a critical review. Biotechnol Biofuels 13, 58 (2020). https://doi.org/10.1186/s13068-020-01697-w

2) Vaidya, A.A., Murton, K.D., Smith, D.A. et al. A review on organosolv pretreatment of softwood with a focus on enzymatic hydrolysis of cellulose. Biomass Conv. Bioref. (2022). https://doi.org/10.1007/s13399-022-02373-9

3) Ostadjoo S, Hammerer F, Dietrich K, Dumont MJ, Friščić T, Auclair K. Efficient Enzymatic Hydrolysis of Biomass Hemicellulose in the Absence of Bulk Water. Molecules. 2019 Nov 20;24(23):4206. doi: 10.3390/molecules24234206. PMID: 31756935; PMCID: PMC6930478. doi: 10.3390/molecules24234206

4) Huang L-Z, Ma M-G, Ji X-X, Choi S-E and Si C (2021) Recent Developments and Applications of Hemicellulose From Wheat Straw: A Review. Front. Bioeng. Biotechnol. 9:690773. doi: 10.3389/fbioe.2021.690773

Round 2

Reviewer 2 Report

The article molecules-1794701 showed improvements, but there are still conceptual errors. In addition, the use of the English language needs to be revised.

I suggest reviewing the cited references one by one. There are inappropriate references in the text.

- Table 1: For 2G correct to “Pretreatments, enzyme hydrolysis  and fermentation [7]. Thermochemical process – direct combustion pyrolysis, gasification”. There are inappropriate references here;

- Table 1: For 3G correct to “Pretreatments, enzyme hydrolysis  and fermentation”;

- Table 1: algae are not raw material for 2G ethanol.

Reviewer 3 Report

The main concerns raised during the first review round have not been clarified by the authors. It is unknown what the purpuse of this work is; moreover, there is a big doubt on the effectiveness of the measures or actions on the increase of sugar yields. For example, in line 182 it is stated that "The high concentration of copper and nickel inhibits the process of photosynthesis, which leads to lower sugar content in Glycine max", whic can constitute just an anecdote rather than a general behaviour.